# Evaluation of Selected Biological and Chemical Treatments against Soil-Borne Diseases of Ginger in High Tunnel Production

**Zelalem Mersha * and Michael A. Ibarra-Bautista**

Agricultural Research Station, College of Agriculture, Virginia State University, 1 Hayden Drive, Petersburg, VA 23120, USA
* Correspondence: zmersha@vsu.edu; Tel.: +1-804-524-2694

**Abstract:** With its multifaceted health benefits, ginger is one of the commonly consumed dietary condiments with continued demand worldwide leading to more importation into countries such as the U.S. The rhizome of ginger, the seed-piece but also the edible portion, is highly vulnerable to pathogen infections causing seed-piece and soil-borne diseases (SSDs). Laboratory and high tunnel (HT) studies were conducted between 2018 and 2020 to evaluate the effect of soil and transplant drenches of selected biocontrol agents (BCAs) and chemicals. In vitro dual-culture tests revealed that the *Trichoderma harzianum* strain T-22 (Th-22) grew significantly faster than *Fusarium oxysporum* f.sp. *zingiberi* (Foz), the fungus causing yellows and rhizome rot in ginger. Th-22 grew more than three times faster than Foz within 3 days and engulfed the pathogen within 7 days when incubated at 30 °C. The bioproducts (RootShield® Plus, LifeGard®), an insecticide/nematicide (Vydate®) and disinfectant (10% Clorox) tested against Foz and any pre-existing pathogens in a HT significantly reduced severity of yellowing and rhizome rot when compared with the non-treated control. Yield was significantly higher for RootShield® Plus compared to the control in one of the trials. Spatially, declining ginger growth but an increased incidence of SSDs were observed as one walked from the door towards the back of the HT. Phospho-lipid fatty acid analysis showed better microbial activity on soils that received drenches of BCAs than on soils sampled pre-treatment.

**Keywords:** ginger rhizome rot; yellows; biological control; soil drench; nematodes

## 1. Introduction

The demand for ginger (*Zingiber officinale* Rosc., Zingiberaceae) has increased sharply in recent years because of its multifaceted health benefits as an herbal medicine. Many of its bioactive components such as gingerol and shogaol [1] are reported to have remarkable health benefits. However, ginger is known to be vulnerable to several fungal, oomycetous, bacterial, viral, and nematode diseases [2–4] affecting quality and quantity of the crop. More importantly, seed-piece and soil-borne diseases (SSDs) limit and threaten its production worldwide [5,6]. For one, there is a lack of germplasm resistance to pathogens causing SSDs owing to the continued use of vegetative means of ginger propagation from the seed-piece rhizome [7]. What also complicates the effect by SSDs is the high water requirement of the crop, which at the same time creates a favorable environment for infection and further spreads harmful pathogens causing these diseases. In regions of the U.S. and other temperate areas, the lack of natural enemies in a new environment compared to the tropical areas where ginger thrives best, and the recurrent use of the same ground in high tunnels, exacerbates the incidence of SSDs. Research that investigates any affordable and eco-benign solution to this problem is a top priority for increased productivity, profitability and adoption of the crop in a new temperate environment.

This study investigates the effect of selected biological and chemical treatments of the production area as part of a mitigation effort against SSDs, mainly yellowing and

rhizome-rot caused by *Fusarium oxysporum* f.sp. *zingiberi* [8] and root knot nematode caused by *Meloidogyne incognita* [9], which were detected in the same high tunnel (HT) where this research was conducted between the years 2018–2020. These two primary pathogens, plus other yet unidentified bacteria and oomycetes, resulted in 5–70% of crop mortality in the high tunnel during the initial assessments in October 2018 [10]. Rhizome root rots, fusarium yellows, bacterial wilts and root knot nematodes inflicted up to 50–90% yield losses in other ginger production areas [11].

Beneficial microbes are amongst the frequently used biocontrol agents (BCAs) for managing many plant diseases in general [12] and to control SSDs caused by *Fusarium* spp. and *Pythium* spp. on ginger specifically [13]. BCAs from the genus *Trichoderma* and *Bacillus* are the most widely used beneficial microbes against soft rot and root rot caused by pathogens on ginger [14]. For example, growth of *Fusarium oxysporum* f.sp. *zingiberi* (Foz) was inhibited by *Trichoderma harzianum* [15–17], *Trichoderma viride* [18] or both species [19,20]. In this study, RootShield® WP (a.i.: *Trichoderma harziaum* Rifai Strain T-22) and RootShield® Plus WP (a.i.: *T. harzianum* strain T-22 and *T. virens* strain G-41), both acquired from BioWorks Inc., Victor, NY, USA, were tested as Trichoderma-based products. LifeGard® (a.i.: *Bacillus mycoides* isolate J), acquired from Certis USA LLC Columbia, was tested as a Bacillus-based biocontrol product. The study started by conducting in vitro mono- and dual-culture experiments thereby generating data that would be useful in determining the rates of growth rates and interactions between *Trichoderma* and Foz.

Efficacy of a wide range of contact and systemic fungicides [21] and disinfectant solutions was studied against yellows and other SSDs of ginger. Mohanty et al. [22] reported best control of nematodes by combining sawdust mulch and Nemacur or Oxamyl (Vydate® L) as post-planting treatment and reduced incidence of *Fusarium* only by post-plant treatment with Nemacur or Oxamyl. Mao et al. [23] also reported Choloropin reducing damage by bacterial wilt when applied as a soil fumigant. Dohroo et al. [24] on the other hand reported hot water treatment at 47 °C for 30 min followed by soil application of *Trichoderma harzianum* and three drenches of mancozeb as most effective in limiting the incidence of soft rot on ginger and in improving the growth and yield. Treatment of infected rhizome, soil-drenching, and synergistic activity of chemicals was also reported to show better results [25]. Oxamyl (Vydate® L) and sodium hypochlorite (10% Clorox, 0.525% NaOCl were tested as a soil drench in this study to compare efficacies with other BCAs. A final aspect covered in this research is elucidating the response of BCAs to a range of microclimatic conditions. This will be pivotal to optimize efficacy of BCAs, in contrast to a wider perception of quick fix, in relation to the range of weather parameters favoring the harmful pathogens.

## 2. Materials and Methods

**Sample collection and pathosystem identification.** In October 2018, ginger plants grown in a high tunnel for demonstration purposes were scouted. A stunted plant was sampled from one of the local pockets among the affected areas and diagnosed for macroscopic and microscopic features of SSDs. Macroscopic features were described after visual observations of discoloration, chlorosis, necrosis or any malformation on the foliage as well as freshly dissected rhizomes. Microscopic features of the respective pathogenic organisms were observed on stereo and binocular microscopes. One of the causative agents, *Fusarium oxysporum* f.sp. *zingiberi* (Foz), was reported [8] after three confirmatory approaches: cultural and microscopic appearances [26], repeated pathogenicity tests, and following a molecular identification technique. For molecular diagnosis, DNA was extracted from conserved regions using four primers, namely internal transcribed spacer (ITS), translation elongation factor (EF), β-tubulin (Bt), and calmodulin (CaM) according to methods outlined by White et al. [27], O'Donnell et al. [28], Glass and Donaldson [29] and Carbone and Kohn [30], respectively. PCR products of all sequences were deposited in GenBank. In addition, swollen parts typical of a root knot nematode were visually observed from plants uprooted in the same high tunnel in 2018 and from another seed-piece rhizome

planted in a standard potting mix in 2019 (Figure 1f). Parts of the samples were mounted microscopically (Figure 1g,h) and sent to the nematology lab in Virginia Tech University in 2019 and the causative agent was identified as *Meloidogyne incognita* [9].

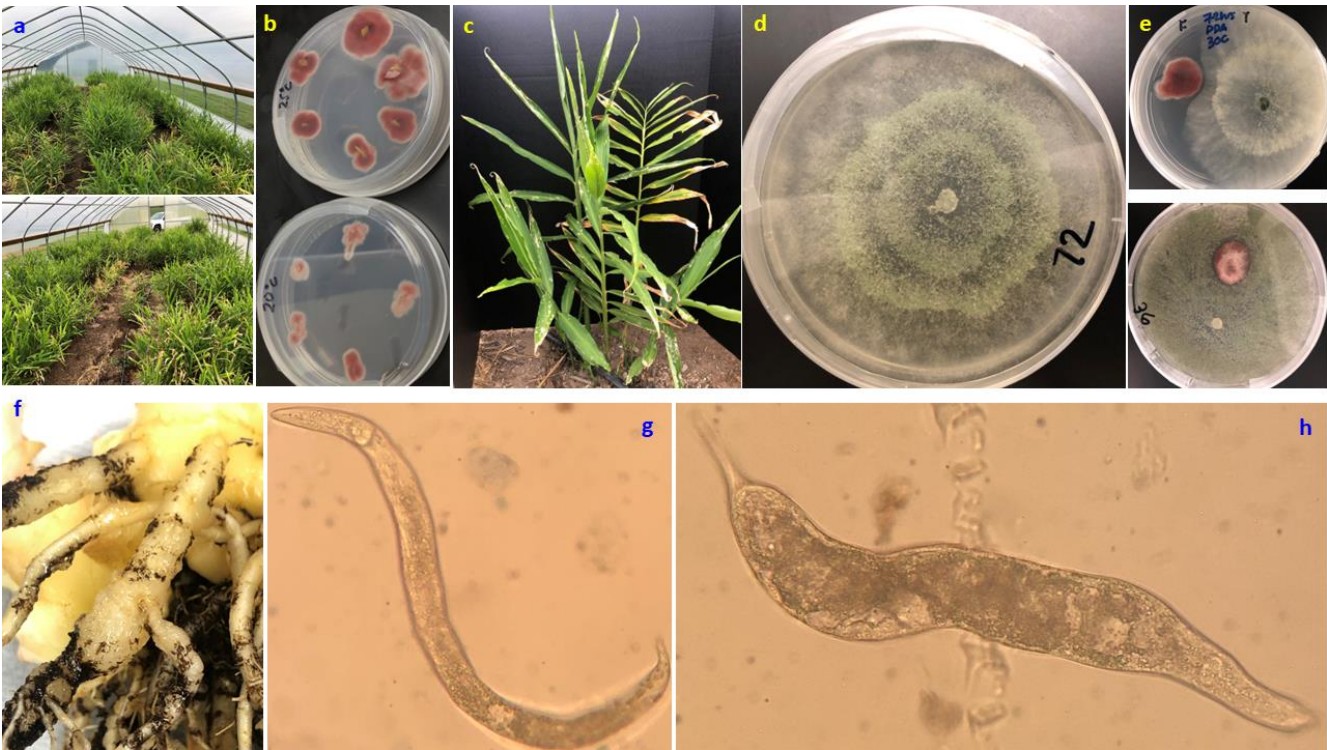

**Figure 1.** (**a**) front (top) and back (bottom) view of a high tunnel with pockets of affected areas, a typical spatial pattern of seed and diseases (SSDs); (**b**,**c**) isolation of *Fusarium oxysporum* f.sp. *zingiberi* and yellowing symptom after artificial inoculation; (**d**,**e**) *Trichoderma harziahum* strain T-22 and dual cultures on potato dextrose agar (PDA) showing the interactions 3 days (top) and 9 days (bottom) after inoculation at 30 °C; (**f**) root knot symptom on ginger; (**g**,**h**) adult and female *Meloidogyne incognita*.

**In vitro experiments.** Mono- and dual-culture tests were conducted in vitro to measure the growth and investigate the interactive effects between the beneficial fungus *Trichoderma harzianum* Rifai strain T-22 (Th-22) and Foz isolate (GFVA3). The in vitro test with complete randomized design (CRD) consisted of five replicates of three factors: (a) two culture media (potato dextrose agar/PDA, acidified PDA), (b) three temperatures (20, 25, or 30 °C), and (c) four treatment combinations, namely TMC (Th-22 monoculture), TDC (Th-22 dual culture), FMC (Foz monoculture) and FDC (Foz dual culture). Confrontational dual culture tests were made by aseptically putting 5 mm plugs of mycelium of Foz and Th-22 (Figure 1d,e). Data on mycelial growth (in 85 mm Petri dish) were collected every 1–3 days. Area under mycelia growth curve (AUMGC) was calculated based on the following formula $AUMGC = \sum_{i=1}^{n}[\left(\frac{(x_i+x_{i+1})}{2}\right) \times (t_{i+1} - t_i)]$. The actual trend of mycelial growth was fitted to a quadratic function and means for the AUMGC were separated according to Tukey's test in SAS version 9.4 (SAS Institute Inc., Cary, NC, USA).

**Trials in High Tunnel.** On 12 June 2019, ginger variety 'Chinese' transplants were raised in 6 inch (15.24 cm) pots for 14 weeks in a temperature-regulated greenhouse and transplanted into the same high tunnel known to have a history of diseases. Holes (2 ft (0.62 m) width × 1 ft (0.31 m) depth) were dug and the following five treatments were spray-drenched on the soil 2–3 weeks before transplanting: (a) RootShield® Plus WP (BioWorks Inc., Victor, NY, USA) at the rate of 24 fl. oz./acre (680.4 g/4046.9 m²), (b) LifeGard™ WG (Certis USA LLC, Columbia, MD, USA) at 4.5 fl. oz. (128 g) per 100 gallons (378.5 L) of water, (c) Clorox 10% solution (0.525% NaOCl), (d) Vydate® L (Corteva, Wilmington, NC,

USA) at 1 gal/A (3.78 L/4046.9 m$^2$), and (e) distilled water as a non-treated control. All five treatments were drenched twice, in and around the rhizosphere area of 3 ft (0.91 m) width by 2 ft depth, 2–3 weeks before transplanting and 3 weeks after transplanting. All soil-drenches were made using a battery-operated Chapin$^®$ (Chapin International Inc., Batavia, NY, USA) backpack sprayer with a total volume of 2 gallons drenched over 12 holes. The research was conducted in a 20 ft (6.1 m) width × 45 ft (13.7 m) length high tunnel with manually operated side vents. The trial with randomized complete block design (RCBD) consisted of five treatments randomized in three blocks, each row considered as a replicate. Four plants spaced 2 ft (0.62 m) apart were used for a single treatment and data were collected from the inner two plants. The three rows were 2.5 ft (0.76 m) away from both wall ends but 5 ft (1.52 m) apart between each.

On 13 July 2020, three varieties, namely 'Chinese', 'Yellow' and 'Bluering' were transplanted and the treatments were slightly modified: (a) RootShield$^®$ Plus soil drench, (b) RootShield$^®$ Plus transplant (a week before transplanting) and soil drench, (c) LifeGard™ WG soil drench, (d) Vydate$^®$ L soil drench, and (e) distilled water soil-drenched as non-treated control. Additionally, 10 mL of $2 \times 10^3$ cfu of the pathogen *Fusarium oxysporum* f.sp. *zingiberi* from the isolate GFVA3 was spray-inoculated in each hole that was readied for the transplants a week before treatments started.

In both trials, disease severity was scouted visually by estimating the extent of yellowing in percent. Likewise, insect pest damage (grasshopper, shoot borer and armyworm caterpillar) was visually rated by comparing the percentage of affected area with the total foliage of the plant. Gall formation on the root was scored at the end of each trial using a root gall index based on a scale of 0 to 10 with zero representing completely healthy with no galls and 10 representing severe (100%) galling according to Zeck [31]. Growth parameters such as total volume of foliage, leaf area measured using LI-COR leaf area meter, weight of foliar biomass, and yield were recorded at the end of the trials on 11 November 2019 and 25 November 2020. Spatial observations of growth parameters were fitted to the exponential decay function and that of the disease parameters to the exponential growth function.

**Microclimate in high tunnel.** Temperature and relative humidity were recorded at 30 min intervals using two TinyTag$^®$ Plus 2 data loggers (Gemini Data Loggers, Chichester, UK) that were placed inside and outside the HT. Hourly data were presented as a trend line and further day-night temperature and relative humidity difference were developed using a pivot table to compute differences between inside and outside the HT by considering 7:00 a.m.–18:59 p.m. and 19:00 p.m.–6:59 a.m. as a day and night duration, respectively.

**Soil PLFA analysis.** In 2020, soil samples representing pre- and post-treatments of selected spray-drenches were sent for phosphor-lipid fatty acid (PLFA) analysis to Microbial ID Inc. (Newark, DE, USA). Soil samples collected from all five treatments in 2020 plus two samples that were collected in 2019 (pre-treatment and oxamyl post-treatment) were kept at 4 °C in a walk-in cooler, freeze-dried and sent for analysis. Samples from these seven treatments were sent in triplicates.

## 3. Results

**Identification of causative agents.** Spatially, an aggregated pattern of either stunted growth or mortality of ginger plants (empty spots) was vividly discernible at the back of the tunnel compared to the front (Figure 1a). Accordingly, incidence of SSDs ranged from 5–70% on the six rows but with a clear increment gradient of the disease incidence as one walked from the door to the back of the HT (Figure 1a). Visually, a clear yellowing symptom on foliage and discoloration on diseased rhizomes were discerned and, upon culturing on PDA, pinkish-colored colony growth was exclusively observed on the backside of the culture media (Figure 1b,c). The isolated fungi grew faster and better when incubated at 25 °C compared to 20 °C (Figure 1b). *Fusarium oxysporum* f.sp. *zingiberi* (Foz) was reported as a causative agent of yellowing and rhizome rot disease in the continental U.S. [8] based on confirmatory studies on symptomatology, morphological appearances [26,28], outcomes

of pathogenicity tests, and submission of sequences in GenBAnk, i.e., accession #MT337417 for ITS, MT436712 for Bt, MT802441 for cal, and MW816632 for EF.

Root knot symptoms and the associated nematodes captured on a binocular microscope are shown in Figure 1f. Parts of the samples showing a typical root knot appearance were mounted microscopically whereby juvenile and female nematodes were observed (Figure 1g,h). The causative agent was identified as *Meloidogyne incognita* [9] from a sample that was sent to the nematology lab at Virginia Tech University. Other yet unidentified species of fungi, bacteria, and water molds have also been detected microscopically at various times during the diagnosis processes starting 2018, but a complete characterization using DNA extraction and pathogenicity test is not complete.

**In vitro experiments.** *Trichoderma harzianum* strain T-22 (Figure 1d) grew faster in monoculture and reached the maximum 85 mm radial growth within 3, 5 and 6 days when incubated at 30, 25 and 20 °C, respectively (Figure 2). The growth of Th-22 was slightly but not significantly delayed in dual cultures when the trend over time (Figure 2) or the areas under the curve of mycelial growth (AUCMG) (Figure 3) were compared with the growth in monocultures. *Fusarium oxysporum* f.sp. *zingiberi* (Foz) grew slightly better in monocultures when incubated at 25 °C and 30 °C than when at 20 °C (Figure 2). At 30 °C, Foz grew significantly higher in monoculture than its growth in dual culture with Th-22. The prediction using the quadratic function was highly significant ($p < 0.0001$) in all cases with a coefficient of determination $R^2$ ranging between 0.856 and 0.997 (Figure 2). The quotients of the linear term from the monoculture in the prediction showed 2.9–4.9 folds of more growth of Th-22 than Foz.

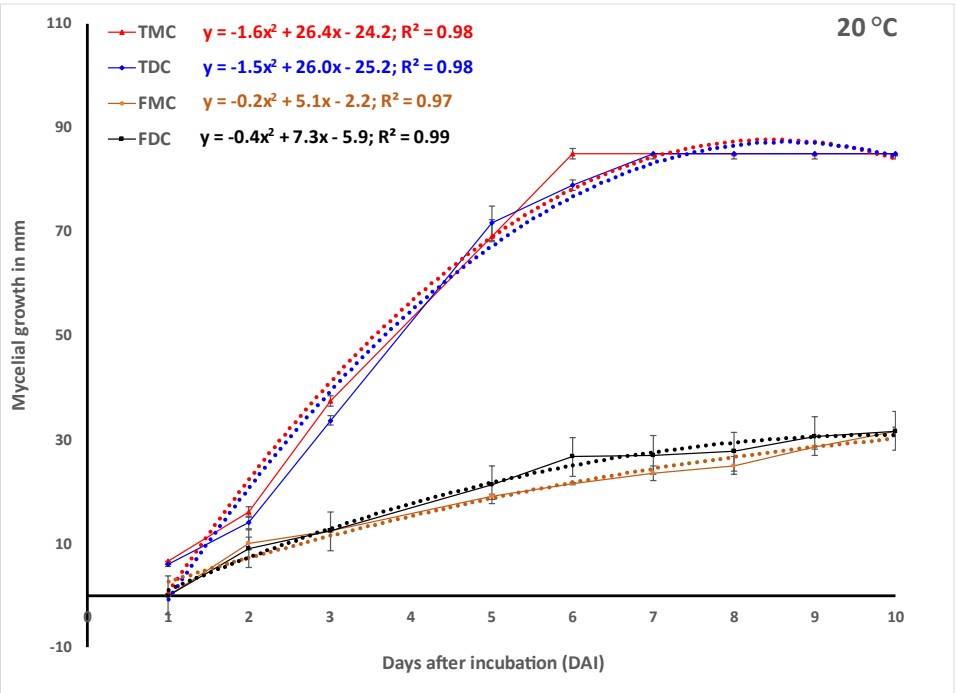

**Figure 2.** *Cont.*

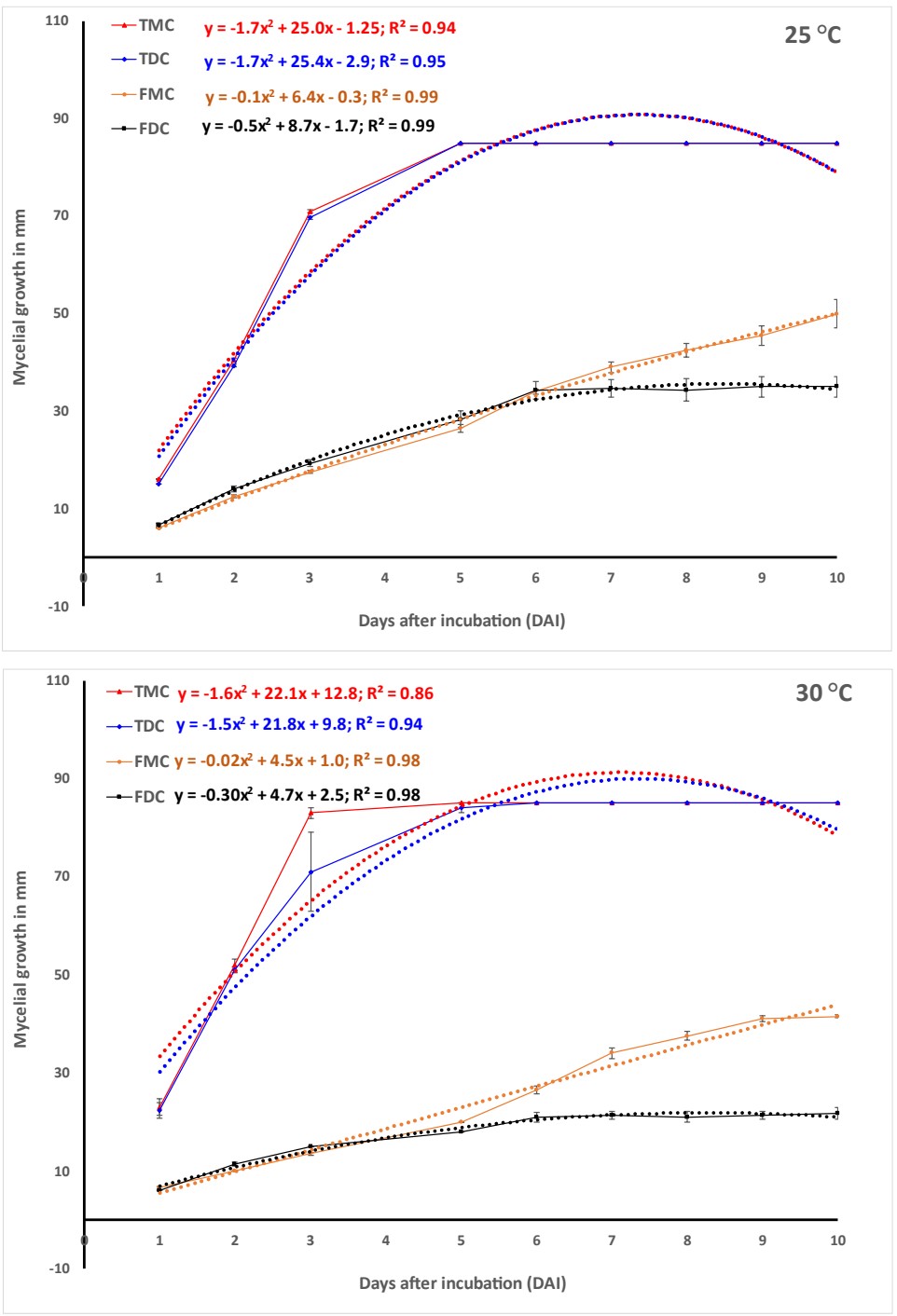

**Figure 2.** In vitro mycelial growth of the pathogen causing yellowing and rhizome rot of ginger *Fusarium oxysporum* f.sp. *zingiberi* and the beneficial fungus *Trichoderma harzianum* strain T-22 in monocultures (FMC, TMC) or dual cultures (FDC, TDC) on potato dextrose agar (PDA) medium incubated at 20, 25 and 30 °C. The continuous lines are the actual data points and the dotted lines represent predicted data according to a second-degree polynomial function.

Foz, on the other hand, grew slower than the beneficial fungus *Trichoderma harzianum* (Th-22) in all cases irrespective of the incubation temperature (Figure 2). The contrast in mycelial growth in the monoculture vs. dual culture of Foz did not show any significant difference (Figure 2). However, the Th-22 grew slightly higher in monoculture than in dual culture. Fusarium, on the other hand, grew slightly less in five monocultures out of the six contrasts made (Figure 2).

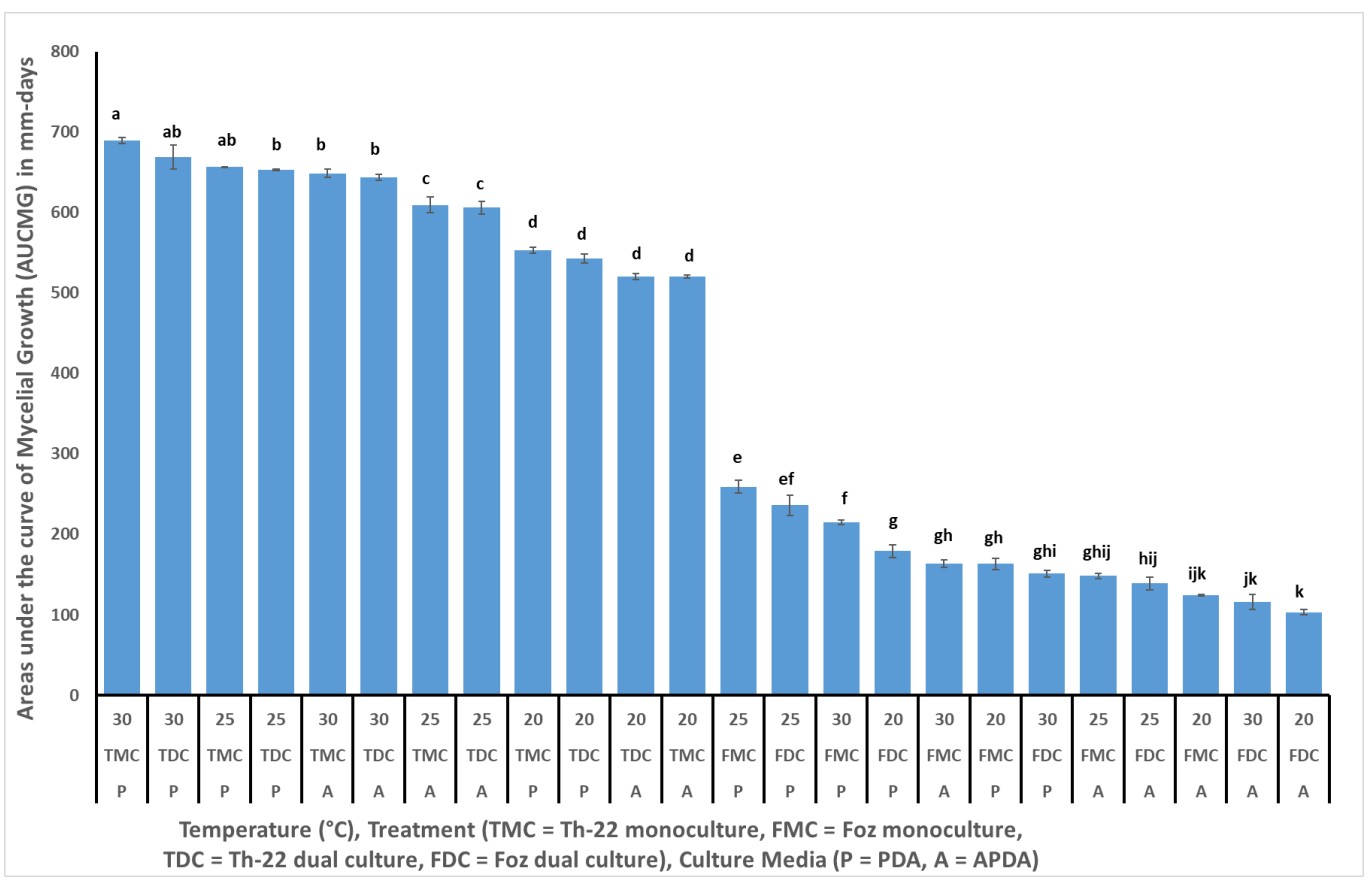

**Figure 3.** Areas under the mycelial growth curve (AUMGC) of the pathogen *Fusarium oxysporum* f.sp. *zingiberi* (Foz) and the beneficial fungus *Trichoderma harzianum* (Th-22) in mono- or dual culture laboratory studies plated on potato dextrose agar (PDA) or acidified PDA (APDA) media and incubated at 20 °C, 25 °C or 30 °C. Result from interaction of three factors, i.e., Temperature × Treatment × Culture Media is presented after a significant *p*-value = 0.0008 using Proc GLIMMIX procedure in SAS. Means that do not share the same letter on top of the bars are significantly different according to Tukey's test at *p* < 0.05.

Treatment means for the AUMGC separated using Tukey's test at *p* < 0.05 level is presented in Figure 3. The trichoderma strain in this study Th-22 grew significantly faster in monoculture and in dual culture than Foz when incubated at 30 or 25 °C in both PDA and APDA (Figure 3). The growth of Th-22, however, was significantly lower when incubated at 20 °C irrespective of the culture media. In contrast, Foz grew significantly lower at all levels of temperature and culture medium and the four treatment combinations when compared with the growth of Th-22 (Figure 3). At 30 °C, percent inhibition of growth 10 days after incubation on Foz was 47.8% and 40.2% on PDA and APDA, respectively (data not presented).

**Disease mitigation studies in the HT**. In this study in 2019, twice soil-drenched RootShield® Plus showed better yield, 66.7% higher than the non-treated control, and resulted in percent protections of 54.1, 47.7, and 19.5 against foliar disease severity (yellowing and stunting), nematode, and insect damages, respectively (Table 1). All soil drenches significantly reduced disease severity (scouted 142 days after planting) and root-knot nematode scoring (at harvest 152 days after planting) when compared to the non-treated control, except the drench, with 10% Clorox which was not significantly different in terms of nematode scoring. Yield of ginger that resulted after RootShield® Plus drenches was significantly higher than the non-treated control but not for the other treatments (Table 1).

**Table 1.** Efficacy of the two biocontrol products, a nematicide/insecticide and a disinfectant soil-drenched in the study to mitigate seed- and diseases on ginger grown in the high tunnel in 2019.

| Treatment | Disease Severity and Stunting (%) 142 DAP * | | Insect Damage (%) 142 DAP | | Root Knot Nematode Scoring at Harvest 152 DAP | | Yield (kg) at Harvest 152 DAP | |
|---|---|---|---|---|---|---|---|---|
| RootShield® Plus WP | 31.87 | b ** | 15.30 | ab | 1.83 | b | 1.10 | a |
| LifeGard® WG | 33.20 | b | 13.67 | b | 1.67 | b | 0.61 | b |
| Vydate® L | 24.87 | b | 11.00 | b | 0.67 | b | 0.71 | ab |
| Clorox 10% | 39.86 | b | 10.17 | b | 2.00 | ab | 0.54 | b |
| Non-treated control | 69.37 | a | 19.00 | a | 3.50 | a | 0.37 | b |
| *p*-value | 0.0325 | | 0.0089 | | 0.0221 | | 0.0273 | |

\* DAP = days after planting; ** Means that do not share the same letter within a column are significantly different according to Tukey's test at $p < 0.05$.

In the trial in 2020, no significant difference ($p < 0.05$) was observed among the three varieties and their interaction with the treatment, and thus only the effects of the treatment were compared. Severity of yellowing and stunting was significantly higher in non-treated plants than treated plants when assessments were made 25 days after planting and when total volume of disease was calculated in terms of AUDPC (Table 2). In this trial, there was no significant difference amongst treatments or varieties in terms of plant height, the leaf area measured using a LICOR meter and yield, thus no data are presented. However, there was a significant difference among the three varieties in terms of foliar biomass measured ($p = 0.0091$) with no significant effects of treatment ($p = 0.7871$) or the interaction between treatment and variety ($p = 0.4853$). The yield, accordingly, was 530.4 g for Yellow, 387.5 g for Bluering and 267.7 g for Chinese varieties. Oxamyl-drenched treatments had a significantly lower disease severity, insect damage and root knot nematode damage, but not yield.

**Table 2.** Efficacy of biocontrol products and chemicals transplanted and soil-drenched in the study to mitigate seed- and diseases on ginger grown in the high tunnel in 2020.

| Treatment | Application Method, Rate | Disease Severity (%) and Area Under Disease Progress Curve | | | | | | Insect Damage | |
|---|---|---|---|---|---|---|---|---|---|
| | | 25 DAP * | | 83 DAP | | AUDPC | | 48 DAP | |
| RootShield® Plus | Soil drench | 2.72 | b ** | 5.56 | b | 366.3 | b | 5.47 | ab |
| RootShield® Plus | Transplant and soil drench | 3.94 | b | 6.64 | b | 403.2 | b | 4.75 | b |
| LifeGard™ WG | Soil drench | 3.95 | b | 4.89 | b | 343.4 | b | 4.91 | ab |
| Vydate® L | Soil drench | 4.28 | b | 8.30 | ab | 470.4 | b | 5.19 | ab |
| Non-treated control | Soil drench | 11.28 | a | 13.00 | a | 895.4 | a | 9.30 | a |
| *p*-value | Treatment (Trt) | 0.0002 | | 0.0027 | | 0.0002 | | 0.0323 | |
| | Variety (Var) | 0.1890 | | 0.0641 | | 0.0734 | | 0.8687 | |
| | Trt × Var | 0.9994 | | 0.7307 | | 0.9836 | | 0.3192 | |

\* DAP = days after planting; ** Means that do not share the same letter within a column are significantly different according to Tukey's test at $p < 0.05$.

Spatially, growth parameters in terms of plant height (cm) and total volume of foliage (cubic meters) showed decrement as one walked from the front door towards the back end of the rows (Figure 4). For instance, average plant height (mean ± SE) within 0.6 m of the door was 77.0 ± 3.5 cm compared to only 52.5 ± 9.8 cm at 12.2 m distance from the door (Figure 4a). The trend was fitted well to and described by the exponential decay function with coefficient of determination values of 0.73 ($p = 0.0037$) and 0.85 ($p = 0.0002$), respectively. On the other hand, disease parameters in terms of severity of yellowness and stuntedness (%) and root knot nematode scoring (0–5 scale) increased as one walked from the front door to the back of the HT. Average percent disease severity (mean ± SE) within 0.6 m of the door was 10.7 ± 2.4 % compared to 52.3 ± 13.5 % at 12.2 m distance from the door (Figure 4c).

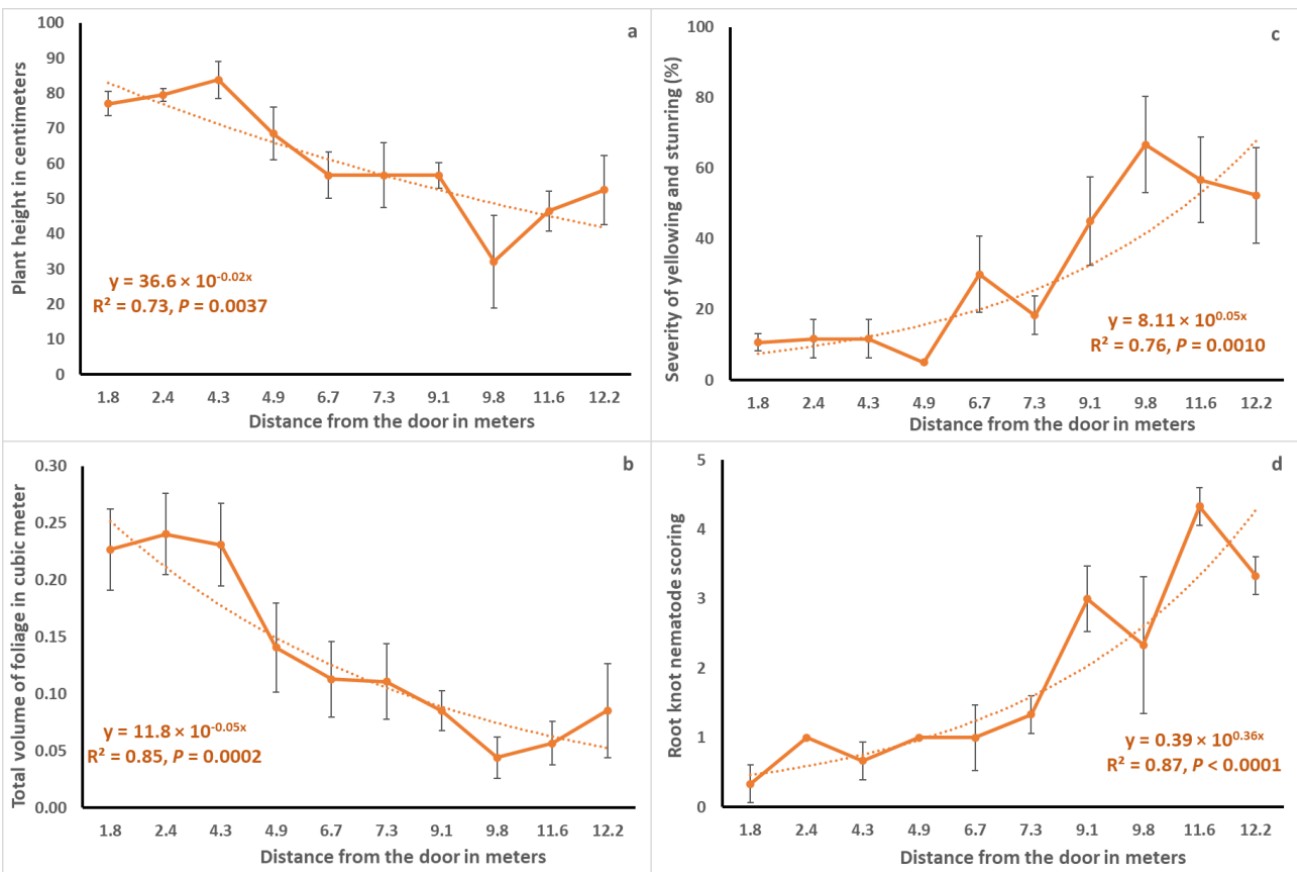

**Figure 4.** Spatial pattern of growth parameters (left) in terms of (**a**) plant height (cm) and (**b**) total volume of foliage (cubic meters) and ginger plant health parameters (right) in terms of (**c**) disease severity (%) and (**d**) root knot nematode rating (0–5) following Zecks's [31] scale as one walks from the door to the back of the high tunnel. The continuous lines are the actual data points and the dotted lines represent predicted data according to the exponential function.

**Microclimate contrast inside and outside the HT**. Generally, temperature was warmer and relative humidity was higher inside the HT compared to the outdoor raised bed where ginger was grown (Figure 5). The only exceptions were 3 days out of a total of 33 days whereby the difference in (inside–outside) temperature resulted in negative values. Even then, the inside temperature was warmer by 2.3 °C when compared with the outside. Relative humidity, however, was exclusively higher inside the HT than outdoors (Figure 5). On average, relative humidity was 46.3% higher inside the HT than outdoors when daily averages were considered for the 33 days' duration during which loggers collected data. A day-to-night contrast of temperature and relative humidity revealed a similar outcome whereby day temperatures were slightly higher than night temperatures when days were

warmer, but the contrast sharply increased on cooler days implying the necessity of high tunnels for season extension when temperatures get lower in late fall and winter. For relative humidity, these proportions were even higher, with 79% and 94% for day and night durations, respectively. Whereas temperature differences were distinctively higher when the days were cooler, relative humidity was clearly higher during most of the days with differences as high as 93% recorded during the month of October. The difference in relative humidity between the inside and outside was 23.5% during the daytime, and 68.0% during nighttime (Figure 5).

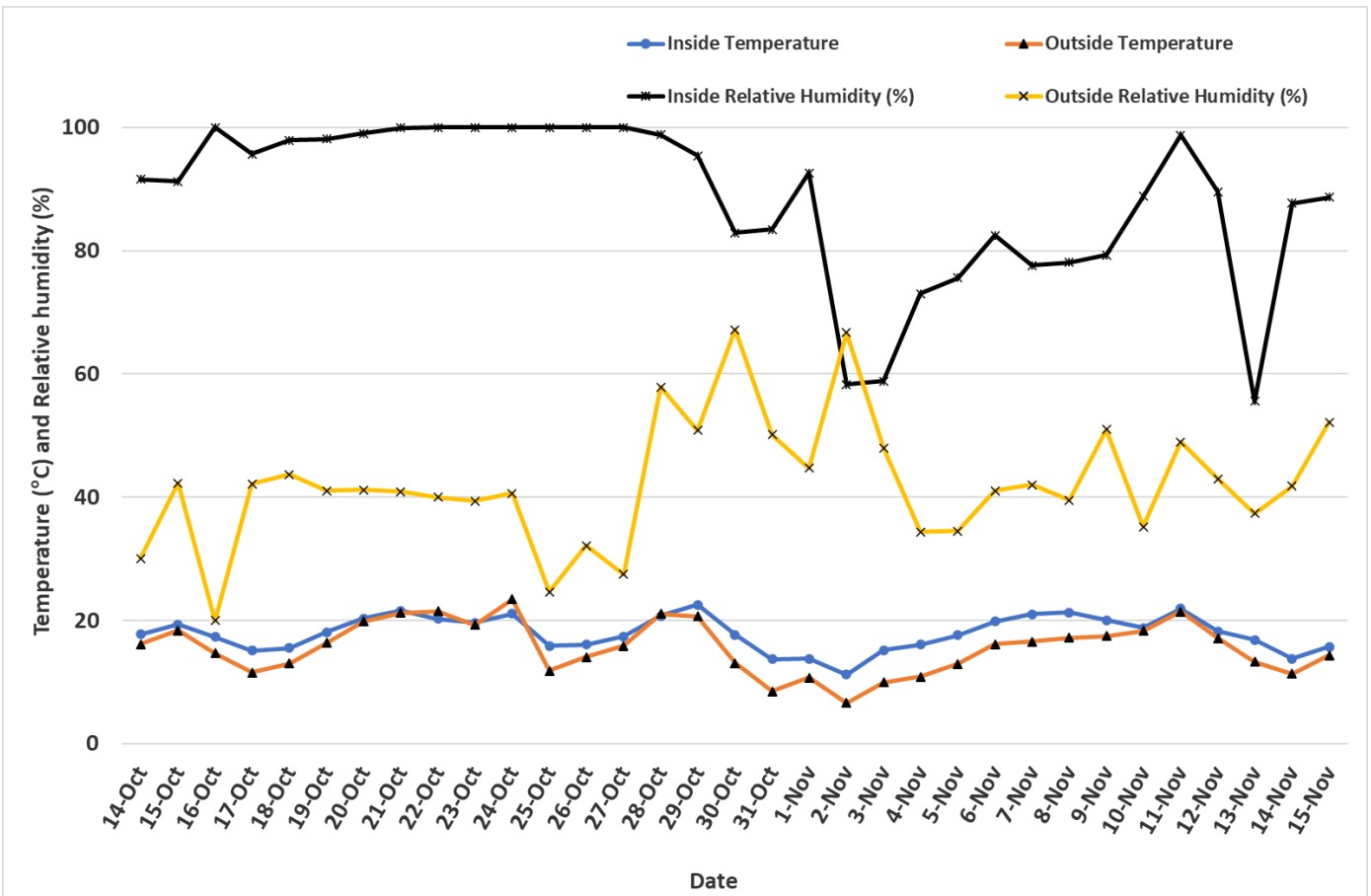

**Figure 5.** Contrast of temperature (°C) and relative humidity (%) inside and outside the high tunnel (HT) where the trial was conducted. A TinyTag® data logger was placed at ground level in the middle of the HT and another one placed on a raised bed outside the HT where ginger plants were grown.

**Soil microbial profile.** Samples sent to Microbial ID Inc. for Phospho-Lipid Fatty Acid (PLFA) analysis showed significantly different outcomes in terms of the various parameters analyzed (Table 3). For instance, pre-plant samples in both years had significantly lower percentages of arbuscular-mycorrhizal fungi (AMF), eukaryotes, and gram-negative bacteria than the post-drenching treatments (Table 3). Total biomass (nanomoles/gram soil) was significantly lower when pre-planting soil samples were compared to the three BCAs in 2020 but not to the samples from the oxamyl or the non-treated control (Table 3). On the other hand, the microbial-type fungal and gram-positive bacterial population were higher in samples collected pre-planting than post-treatment collected samples for both years.

**Table 3.** Microbial population in soil samples collected pre- and post-treatment studies in the high tunnel.

| Sample Source (Drench) | Microbial Type (%) | | | | | | Total Biomass (nmole/g) |
|---|---|---|---|---|---|---|---|
| | Fungi | Arbuscular Mycorrhizal Fungi (AMF) | Eukar-yotes | Actino-mycetes | Gram Positive Bacteria | Gram Negative Bacteria | |
| **Samples from 2019** | | | | | | | |
| Pre-plant | 5.11 ab * | 2.06 c | 0.34 c | 12.24 b | 43.14 a | 37.1 d | 65.0 cd |
| Oxamyl (soil) | 2.91 b | 3.43 b | 1.62 b | 12.21 b | 35.17 b | 45.3 a | 67.6 bcd |
| **Samples from 2020** | | | | | | | |
| Pre-plant | 6.11 a | 1.95 c | 0.43 c | 12.21 b | 42.08 a | 37.2 d | 59.0 d |
| Th + Tv (plant and soil) | 2.58 b | 5.50 a | 2.35 ab | 14.24 a | 33.24 b | 42.1 b | 87.9 a |
| Th + Tv (soil) | 2.35 b | 5.52 a | 2.33 ab | 14.78 a | 34.96 b | 40.1 c | 76.8 ab |
| *Bm* − J (soil) | 2.77 b | 5.73 a | 2.54 a | 14.03 a | 33.78 b | 41.1 bc | 83.9 ab |
| Oxamyl (soil) | 2.18 b | 5.57 a | 2.52 a | 14.94 a | 34.60 b | 40.19c | 74.1 abcd |
| Non-treated (soil) | 2.26 b | 5.21 a | 2.20 ab | 15.15 a | 34.66 b | 40.52bc | 71.8 abcd |
| ***p*-value** | **0.082** | **0.001** | **0.001** | **0.001** | **0.001** | **0.001** | **0.014** |

* Means that do not share the same letter within a column are significantly different according to Tukey's test at $p < 0.05$.

## 4. Discussion

Seed- and soil-borne diseases such as yellowing and rhizome rots caused by fungal, bacterial and oomycetous microbial pathogens, as well as infections by nematodes, gravely affect productivity of ginger worldwide. Among these, fusarium yellows and rhizome rot caused by *Fusarium oxysporum* f.sp. *zingiberi* are the most common species and responsible for major loss in ginger production by decaying ginger rhizomes [5,6]. This study presented investigations undertaken to partly identify two important pathogens that were prevalent on ginger in the study area in Virginia, and reported outcomes of high tunnel studies that were conducted from 2018–2020 and geared towards comparing efficacy of microbial beneficials and chemical drenches.

A study in Queensland, Australia, attributed poor establishment of ginger to be caused primarily by *F. oxysporum* f.sp. *zingiberi* (Foz) and synergistically with the soft rot disease caused by *Erwinia chrysantheni* [5]. Soft rot primarily caused by *Pythium* spp., yellows caused by *Fusarium* spp., bacterial wilt and leaf spots have also been reported as major diseases of ginger in India [32] and around the world. Yellowing and rhizome rot caused by Foz [8] and root knot nematode caused by *Meloidogyne incognita* [9] were identified in Virginia, also causing a similar poor stand in the high tunnel ginger production. The spatial gradient of disease increase as one walked towards the back of the tunnel during the study durations in 2019–2020 also significantly corroborated the initial observation of pocketed mortality and stunted growth observed at the beginning of the study in 2018.

The four treatments, namely the two biological control products RootShield® Plus and LIfeGard®, provided 54–59 and 52–61 percent protections, respectively, against the SSDs when disease severity of each of these treatments was compared with the non-treated control. Similarly, percent protection that was achieved due to the application of the two chemicals Vydate® L and NaOCl was 48–64 and 43.0, respectively. Because of its highly promising effect and wider use of the Trichoderma-based product RootShield® Plus as a seedling dip, a new treatment encompassing two drenches, one on the transplant and another on the soil, was added in 2020. On the other hand, Clorox was dropped in 2020 for its relatively low efficacy, presumably because of the vaporization of chlorine after a drench and also for the lack of space to accommodate a sixth treatment in the high tunnel. Future studies on Clorox will focus on disinfecting the ginger seed-piece as has been highlighted by Nelson [33].

Shanmungam et al. [20] reported a 45.9 and 45.3% reduction of yellows and rhizome rot, respectively, from a field experiment where strain mixture of rhizobacteria and

*Trichoderma harzianum* was applied. The yield gained due to the soil-drenching from the above-mentioned four treatments from the 2019 trial was 66.7% (RootShield® Plus), 22.2% (LIfeGard®), 31.3% (Vydate® L), and 15.6% (NaOCl), respectively. There was no significant difference between treatments in terms of yield in 2020; hence, no data are presented. Among the varieties though, there was a significant difference in terms of total foliage biomass but not in terms of yield per plant, despite the numerical difference between yellow (3.71 lb) (1.68 kg), Bluering (3.40 lb) (1.5 kg) and Chinese (2.94 lb) (1.33 kg).

Jensen et al. [34] reviewed the four modes of action underlying biological control of plant diseases as competition for resources, antibiosis, hyper-parasitism, and induced resistance. The in vitro experiments in this study clearly affirmed the inhibitory effect (upto 47.8% under 30 °C incubation on PDA) and the competitive advantage of Th-22 in outgrowing the pathogen *Fusarium oxysporum* f.sp. *zingiberi* irrespective of the three temperatures (20, 25 and 30 °C) and the two culture media (PDA and APDA) tested in this study. This implies the wide range of favorable temperatures under which Th-22 can thrive and succeed in countering the advancement of pathogens such as Foz, root knot nematodes and others from causing SSDs. One of the species in RootShield® Plus, *Trichoderma virens*, is reported to also mitigate root knot disease on chickpeas [35]. Particularly in temperate areas, where fall and winter temperatures are low outside, growers can potentially apply Th-22 at the end of the season in protected cultivation systems if they ensure complete closure of their high tunnels and greenhouses. As witnessed from the inside-outside temperature contrast, enough warmth of the soil to promote the biological product can be achieved in these structures as long as the doors, side vents and exhaust windows are closed properly. The significantly higher growth of Foz when in monoculture than in dual-culture with Th-22 at 30° hints at additional modes of action by the beneficial fungus which have been extensively discussed in other studies.

The study suggested the extent of disease suppression achieved by the drench applications of the biocontrol products and the chemicals tested in the high tunnel. However, some of the disease, ginger growth, and yield parameters did not show any significant difference from the high tunnel trials. Part of this is attributed to the pre-existing (up to year 2018) inoculum/disease gradient that most likely continued and persisted during the current trials in 2019 and 2020 despite tillage at the beginning of each season. The spatial gradient of decreased ginger growth activity but increased disease severity, explained very well by the exponential decay and growth functions, was a clear indicator for the most likely variability by the pre-existing factors.

The impact of BCAs and chemical drenches on the microbial dynamics between the pre-treatment and post-treatment of drenches hinted at some differences between the pre-planting and post-drench treatment samples. As with the disease and growth gradient, the pre-existing population of microflora and fauna may have been reflected in results obtained after the analysis. Whereas the study hinted at existing differences and future areas of improvement, similar studies that pinpoint the relationship between Pathogens–Biological Control Agents–Soil Health parameters will be very useful to enumerate efficacies of BCAs in the short- and long term. Ongoing research looking into seed-piece treatment with hotwater, biological treatments, and use of membrane-enhancing compounds such as chitosan [36], alone or in combination, will be described in future reports.

**Author Contributions:** M.A.I.-B., undergraduate student in the College of Agriculture at Virginia State University, has been working on ginger disease management projects. He contributed to formulating and editing various parts of the manuscript holistically. Z.M., Virginia State University, has formulated the research hypothesis, analyzed the data and written the manuscript. All authors have read and agreed to the published version of the manuscript.

**Funding:** This research received no external funding.

**Data Availability Statement:** Not applicable.

**Acknowledgments:** Authors acknowledge the following people for technical support and assistance: Kyle T. Claye (VSU student), Addison Caldwell (VSU student College of Agriculture), and Shilpi Chawla (former postdoctoral researcher). Reza Rafie is greatly appreciated for providing ginger transplants and the high tunnel for this study.

**Conflicts of Interest:** The authors declare no conflict of interest.

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
