# Peer review of "Evaluation of Selected Biological and Chemical Treatments against Soil-Borne Diseases of Ginger in High Tunnel Production"

_horticulturae, doi:10.3390/horticulturae8100870_

Round 1
Reviewer 1 Report
Ginger is one of the commonly consumed dietary condiments with multifaceted health benefits, however the rhizome of ginger is highly vulnerable to pathogen infections causing seed-piece and soil-borne diseases (SSDs). Laboratory and high tunnel (HT) studies were conducted between 2018-2020 to evaluate effect of soil and transplant drenches of selected biocontrol agents (BCAs), mainly Trichoderma harzianum strain T-22 (Th-22), and chemicals in this manuscript, which provide some meaningful results and a reference for related further study. However, some obvious mistakes need to be revised throughout the manuscript.
1. In line 208-210, “Trichoderma strain in this study (Th-22) grew significantly faster in monoculture or in dual culture than Foz when incubated at 30 or 25°C in both PDA and APDA was significantly (Fig. 3).” Was this sentence highlighted with yellow wrong?
2. In line 324-326, “The in vitro experiments in this study clearly affirmed the inhibitory effect (upto 325 47.8% under 30°C incubation n PDA)”. Was there anything missing for the letter highlighted with yellow? Was it “on”?
3. The abbreviation of double culture in Fig. 3 (TF/FT) did not coincide with the expression of double culture in lines 103-104 of experimental materials and methods as well as Fig. 2, which were TDC/FDC.
4. In Figure 3, error bars should be added to the data. The results of Fig. 4 are not described in the text. In line 261-262, the content described here should be from Figure 5, not Figure 4.
5. The discussion part is not deep enough, and the reasons for some experimental results are not discussed. For instance, the experimental design in 2019 was different from 2020, what were the reasons and the potential effects on experimental results should be analyzed. Also, the reason why Clorox 10% treatment was not continued should be explained. In “Microclimate contrast inside and outside the HT” paragraph, what are the effects of the difference between internal and external microclimates? What are the impacts on the disease development and yield of ginger? These issues should be discussed otherwise this paragraph is less relevant to the content of the manuscript.
Author Response
Authors response to reviewer 1
Ginger is one of the commonly consumed dietary condiments with multifaceted health benefits, however the rhizome of ginger is highly vulnerable to pathogen infections causing seed-piece and soil-borne diseases (SSDs). Laboratory and high tunnel (HT) studies were conducted between 2018-2020 to evaluate effect of soil and transplant drenches of selected biocontrol agents (BCAs), mainly Trichoderma harzianum strain T-22 (Th-22), and chemicals in this manuscript, which provide some meaningful results and a reference for related further study. However, some obvious mistakes need to be revised throughout the manuscript.
- In line 208-210, “Trichoderma strain in this study (Th-22) grew significantly faster in monoculture or in dual culture than Foz when incubated at 30 or 25°C in both PDA and APDA was significantly(Fig. 3).” Was this sentence highlighted with yellow wrong?
Thank you for pointing this mistake from our end. The statement is corrected in the revised version.
- In line 324-326, “The in vitro experiments in this study clearly affirmed the inhibitory effect (upto 325 47.8% under 30°C incubation nPDA)”. Was there anything missing for the letter highlighted with yellow? Was it “on”?
Thank you for pointing this editorial error, its fixed in the revised version.
- The abbreviation of double culture in Fig. 3 (TF/FT) did not coincide with the expression of double culture in lines 103-104 of experimental materials and methods as well as Fig. 2, which were TDC/FDC.
Thank you for pinpointing this inconsistency in acronym usage. All acronyms are standardized as TMC, FMC, TDC, and FDC across the board in the revised version.
- In Figure 3, error bars should be added to the data.
Thank you for pointing this. We felt addition of SE bars may not necessarily provide additional information rather be redundant and we weren’t sure if it is a requirement since we already presented the three factor interaction analysis of variance using Proc Glimmix (p = 0.0008). The mean separation at Tukey’s test (p < 0.05) already shows the significant differences between treatments. Since this is brought up as a necessity, however, the SE bars are now added to Fig. 3.
The results of Fig. 4 are not described in the text.
Thank you for pointing this. Fig. 4 is now explained in the results section.
In line 261-262, the content described here should be from Figure 5, not Figure 4.
Thank you, the correct figure number is edited accordingly.
- The discussion part is not deep enough, and the reasons for some experimental results are not discussed.
For instance, the experimental design in 2019 was different from 2020, what were the reasons and the potential effects on experimental results should be analyzed. Also, the reason why Clorox 10% treatment was not continued should be explained.
Thank you for looking into this fundamental question. The discussion is now expanded to touch key elements of our findings. When it comes the modifications we made in successive years, we opted to diversify the factors within the available plot size in the high tunnel by selecting better performing treatments from year 1. The study would like to zoom into the prospects of organic options particularly the use of Trichoderma-based and Bacillus-based bioproducts along with some standard insecticide/nematicide chemistries such as oxamyl. In principle, all treatments except Clorox were repeated in 2020. As a common disinfectant, NaOCl2 may have a reduced efficacy when drenched because of vaporization of chlorine. For that reason, we thought of omitting this treatment from soil-drenches during the repeatition in 2020 but expand multiple uses of the Trichoderma-based product. NaOCl2 is under test on seed-piece treatment options.
In “Microclimate contrast inside and outside the HT” paragraph, what are the effects of the difference between internal and external microclimates? What are the impacts on the disease development and yield of ginger? These issues should be discussed otherwise this paragraph is less relevant to the content of the manuscript.
Thank you for the comment. The microclimate data is presented to showcase how relative humidity and temperature vary between the outside and inside protected systems. It may be less relevant to this specific research but it has implications in application of biocontrol agents and in managing the associated pathosystems in confined environments. Authors expanded the interpretation and take-home message from the microclimate information.
Sincerely,
Zelalem Mersha

Reviewer 2 Report
The Article "Evaluation of selected biological and chemical treatments against soilborne diseases of ginger in high tunnel production" by Mersha and Ibarra-Bautista reports data on the activity of commercial formulations based on microbial antagonists and chemical products in the containment of damage from Fusarium oxysporum f. sp. zingiberi and Meloidogyne incognita on ginger. The results of this work show a better effectiveness when the production takes place in a high tunnel.
However, the following major revisions are recommended.
In the first place, in the Materials and Methods, no reference is made to the techniques used for the "identification of causative agents", as instead discussed in the Results. The methodology used should be better defined, distinguishing the techniques for the isolation of microorganisms and the macro- and microscopic observations (with photomicrographs), from those for nematodes.
Furthermore, Table 1 refers to the percentage of damage caused by insects, but it is not clear what type of damage they are, nor how the same percentage is calculated.
With regard to fig. 5, it would be advisable to replace it with a table with average, minimum and maximum temperatures and humidity (for the periods considered), inside and outside the high tunnel.
In the Article, weights, volumes and dimensions should be expressed in SI Units.
The following minor revisions should also be made.
Page 2, line 53: biocontrol agents (BCAs);
55: Trichoderma and Bacillus italics;
69: Fusarium italics;
Page 4, line 148: pre- and post-;
156: the headings of the sub-chapters in the Results are in italics, unlike the Materials and Methods;
Page 5, line 181: Trichoderma harzianum italics (in the article there is confusion between T22, Th-22, Trichoderma spp., It is advisable to find a unique way to indicate the antagonist);
196 and 199: spp no ​​italics;
208: the antagonistic strain in this study (Th-22);
Page 6, line 220: and Trichoderma harzianum (Th-22) in mono or ...
232: in 2019;
Page 7, line 236: compared. Severity of ...
240, Li-Cor, all caps as page 3, line 139 ?;
243-44 Yellow ... Blue ring ... Chinese varieties
Page 8, lines 261-266: these cultural notes are very well known for ginger;
Page 11, line 326: incubation on PDA.
Author Response
Authors response to reviewer 2
The Article "Evaluation of selected biological and chemical treatments against soilborne diseases of ginger in high tunnel production" by Mersha and Ibarra-Bautista reports data on the activity of commercial formulations based on microbial antagonists and chemical products in the containment of damage from Fusarium oxysporum f. sp. zingiberi and Meloidogyne incognita on ginger. The results of this work show a better effectiveness when the production takes place in a high tunnel.
However, the following major revisions are recommended.
In the first place, in the Materials and Methods, no reference is made to the techniques used for the "identification of causative agents", as instead discussed in the Results. The methodology used should be better defined, distinguishing the techniques for the isolation of microorganisms and the macro- and microscopic observations (with photomicrographs), from those for nematodes.
Thank you for the comment. We have now added significant references, which were the basis for the macroscopic features, microscopic observations and molecular characterizations of the Fusarium oxysporum f.sp. zingiberi that is reported in Plant Disease Journal. The reason for the less emphasis on the details was primarily because the content was presented in the publication. Detailed steps for identifying the root knot nematode were taken care in the Nematology Laboratory and the senior nematologist that identified the nematode species is highlighted as personal communication.
Furthermore, Table 1 refers to the percentage of damage caused by insects, but it is not clear what type of damage they are, nor how the same percentage is calculated.
Thank you for the comment and we apologize for not including that piece of information on the M&M. Insect pest damage is visually scouted on a percentage basis and now included in the revised version.
With regard to fig. 5, it would be advisable to replace it with a table with average, minimum and maximum temperatures and humidity (for the periods considered), inside and outside the high tunnel.
Thank you for the suggestion on a preferred presentation of the microclimatic data. Whereas the complexity and the crowdedness of our graphical presentation can be well received, we redacted the data to a daily average (from an every 30 min record) and presented it graphically to show the continuity of the data.
In the Article, weights, volumes and dimensions should be expressed in SI Units.
Thank you for pointing the discrepancy and we apologize for the oversight. All units are now converted to SI Units.
The following minor revisions should also be made.
Thank you for meticulously going through the manuscript and suggesting these editorial suggestions. We made all corrections in the revised version. We maintained Th-22 as acronym of the entire organism and isolate we used in the study Trichoderma harzianum isolate T-22 (Th-22). The acronym T-22 that was on the label of the product to represent the unique isolate is still kept.
Page 2, line 53: biocontrol agents (BCAs);
55: Trichoderma and Bacillus italics;
69: Fusarium italics;
Page 4, line 148: pre- and post-;
156: the headings of the sub-chapters in the Results are in italics, unlike the Materials and Methods;
Page 5, line 181: Trichoderma harzianum italics (in the article there is confusion between T22, Th-22, Trichoderma spp., It is advisable to find a unique way to indicate the antagonist);
196 and 199: spp no ​​italics;
208: the antagonistic strain in this study (Th-22);
Page 6, line 220: and Trichoderma harzianum (Th-22) in mono or ...
232: in 2019;
Page 7, line 236: compared. Severity of ...
240, Li-Cor, all caps as page 3, line 139 ?;
243-44 Yellow ... Blue ring ... Chinese varieties
Page 8, lines 261-266: these cultural notes are very well known for ginger;
Thank you for pointing this. Whereas such microclimate information may be known in ginger, it has implications in the overall disease management perspectives for many specialty crops hence authors feel positive presenting the result. If there is a strong feeling to cut it out, that could be optional in future revisions.
Page 11, line 326: incubation on PDA.
Sincerely,
Zelalem Mersha

Round 2
Reviewer 1 Report
The authors have corrected the mistakes and addressed the issues which are needed to pay attention. To me, I think the present manuscript might be acceptable.
Author Response
Dear esteemed reviewer,
Thank you for volunteering your time reviewing our manuscript. We have gone through it and made few minor editorial corrections which will hopefully improve the english language and writing style of the journal.
Sincerely,
Zelalem Mersha
Corresponding author
Reviewer 2 Report
The proposed changes and suggestions have been correctly accepted. It is only advisable to express weights, volumes and dimensions both in SI Units and as initially reported. For example: 100 gallon (378.5 liters); 2.5 ft (0.76 m), etc.
Author Response
Esteemed reviewer,
Thank you for devoting your time reviewing our manuscript. As per the suggestion, weights, volumes and dimensions are now expressed using the US standard measurement system as in the original submission and metric system of measurement indicated in parenthesis.
Sincerely,
Zelalem Mersha
Corresponding author